# Cost-Aware Contrastive Routing for LLMs

**Reza Shirkavand**
Department of Computer Science
University of Maryland - College Park
rezashkv@cs.umd.edu

**Shangqian Gao**
Department of Computer Science
Florida State University
sgao@cs.fsu.edu

**Peiran Yu**
Department of Computer Science and Engineering
University of Texas at Arlington
peiran.yu@uta.edu

**Heng Huang**[*]
Department of Computer Science
University of Maryland - College Park
heng@cs.umd.edu

## Abstract

We study cost-aware routing for large language models across diverse and dynamic pools of models. Existing approaches often overlook prompt-specific context, rely on expensive model profiling, assume a fixed set of experts, or use inefficient trial-and-error strategies. We introduce Cost-Spectrum Contrastive Routing (CSCR), a lightweight framework that maps both prompts and models into a shared embedding space to enable fast, cost-sensitive selection. CSCR uses compact, fast-to-compute *logit footprints* for open-source models and *perplexity fingerprints* for black-box APIs. A contrastive encoder is trained to favor the cheapest accurate expert within adaptive cost bands. At inference time, routing reduces to a single $k$-NN lookup via a FAISS index, requiring no retraining when the expert pool changes and enabling microsecond latency. Across multiple benchmarks, CSCR consistently outperforms baselines, improving the accuracy–cost tradeoff by up to 25%, while generalizing robustly to unseen LLMs and out-of-distribution prompts.

## 1   Introduction

After a burst of reinforcement-learning and specialized finetuning, the Large Language Model (LLM) [65, 66, 6, 86, 17] ecosystem has fractured: code models excel at generating code but hallucinate outside programming contexts, math-tuned variants solve AIME [46] yet mishandle open-ended dialogue, and instruction chatbots trade being precise for fluency. Production systems therefore host a pool of models with different sizes, licenses, and domain strengths, and decide at run time which one to call for every user prompt, or worse: burden the users with picking the model they need.

A **router** (a.k.a. model selector, mixture-gate) adjudicates that choice online. It dynamically selects the most appropriate LLM from a pool for each input. Without it, users either over-pay by defaulting to the largest model or risk quality regressions by selecting cheaper ones. Current predictive routers [33, 32, 51, 76] for a pool of LLMs fall into two broad camps: parametric routers and non-parametric ones.

Parametric routing methods, such as softmax-based classifiers [24], optimize exclusively for top-1 accuracy without explicit consideration of inference costs. Consequently, they tend to default to selecting expensive models and require full retraining whenever new models are introduced.

---
[*]This work was partially supported by NSF IIS 2347592, 2348169, DBI 2405416, CCF 2348306, CNS 2347617, RISE 2536663.

39th Conference on Neural Information Processing Systems (NeurIPS 2025).

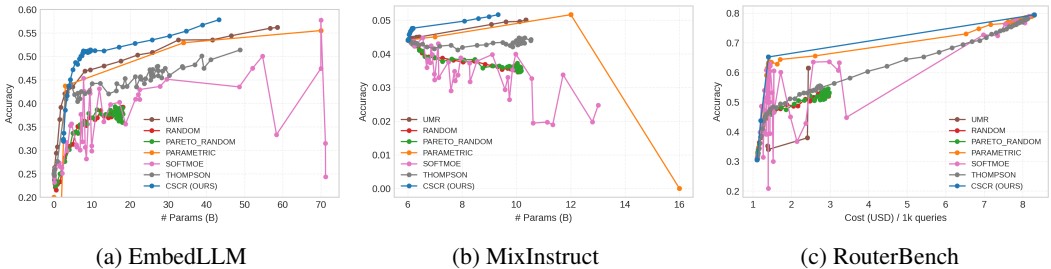

| (a) EmbedLLM | (b) MixInstruct | (c) RouterBench |

Figure 1: **Accuracy–cost/size deferral curves on three expert pools.** Across all benchmarks, our Cost-Spectrum Contrastive Router (blue) consistently dominates the Pareto frontier, achieving higher accuracy at lower model size and latency (left, middle) and reduced cost (right).

Recent non-parametric approaches, such as UMR [40], enhance generalization by routing across a joint space of prompt clusters and model footprints. On the other hand, bandit-based methods [61, 34, 48], including Thompson sampling [85, 2] and UCB [3], track simplified quality–cost metrics that disregard detailed prompt characteristics, resulting in slower convergence and reduced effectiveness, especially when facing heterogeneous and diverse input distributions. All these methods remain cost-agnostic during training, solely relying on post-hoc hyperparameter tuning to achieve an effective balance between cost and accuracy.

In our view, routing boils down to similarity search. If we can embed both prompts and experts in one metric space where cosine distance trades off {quality, cost}, routing reduces to a microsecond Nearest Neighbor query — no brittle softmax gate, no retraining when the pool changes.

In this paper we introduce:

- **Universal ultra-compact descriptors:** Two lightweight and fast to compute fingerprints that work across the full spectrum of LLMs. (1) Logit footprints that require only <10 forward passes through an open-weights model. (2) Perplexity fingerprints that score any black-box API with a small public LM, enabling vendor-agnostic routing.

- **Cost-Spectrum InfoNCE:** A novel contrastive objective that (1) selects correct positives within adaptive cost bands, (2) temperature-scales each band separately, and (3) down-weights negatives in proportion to their cost. This aligns the learned metric with the accuracy–cost Pareto frontier.

- **Routing Efficiency:** A shared space turns routing into a single $k$-NN lookup, eliminating brittle softmax gates and retraining whenever the pool changes. Then a lightweight FAISS index makes routing a microsecond lookup.

- **Comprehensive Evaluation:** We evaluate our method on three routing benchmarks spanning both open-source checkpoints and proprietary APIs. It achieves up to 25% higher accuracy–cost efficiency on a fixed pool of LLMs and demonstrates strong robustness to unseen models and out-of-distribution prompts at inference time.

## 2 Related Work

### 2.1 LLM Routing

**Non-Predictive Routing.** Non-predictive methods generate outputs from one or more models before making a selection. FrugalGPT [10] uses a sequential strategy and a response quality threshold to minimize cost. Other works adopt layered inference architectures to escalate hard queries to more powerful models [91], or leverage cascades with self-verification [54, 98, 43, 51].

**Predictive Routing.** In contrast, predictive routing aims to select the best model *before* any inference is performed. Strategies include supervised learning [76], reward-model-based routing [32], and meta-models trained to predict LLM performance given an input [71]. Router models vary widely in implementation, including neural networks [19, 88, 11, 1], $k$-nearest neighbors [34, 76, 80, 43], matrix factorization [62, 105, 48], and graph neural networks [25]. Others incorporate model-specific tokens or train across multiple domains [18, 7].

Academic routers usually assumed the expert set is static. Recently routing with a dynamic pool of experts has been explored [40, 48]. UMR [40] clusters probes and stores coarse capability footprints but still rebuilds them offline whenever the model pool changes. LLM-Bandit [48] optimizes cost but ignores prompt semantics and offers no cold-start prior for unseen experts.

## 2.2 Routing within MoE and Hybrid Architectures

Routing LLMs can be viewed as a coarse-grained MoE, where each expert is a full LLM. Routing is a central mechanism in MoE models [36, 41, 75], where expert modules are dynamically activated based on input. While classical MoEs involved equally-sized sub-models, modern approaches like Switch Transformer [24] and Mixtral [38] employ sparse activation to minimize cost. Approaches like UltraFuser [20] highlight recent advances in combining model specialization and flexibility.

## 2.3 Model Fusion, Merging and Cascading

Fusion strategies synthesize outputs from multiple LLMs to improve output quality [68, 39, 29, 89, 52]. Fusion approaches often rely on unsupervised metrics [99, 73, 97] or ensemble voting to determine the final output [44]. A related but distinct technique is model merging [50], where weights from multiple pre-trained or fine-tuned models are combined, either directly via methods like weight averaging [92], Task Arithmetic [35], or Fisher merging [56]. In contrast, cascading invokes models sequentially (often ordered by computational cost) and halts once a satisfactory output is generated [10, 98, 30].

To the best of our knowledge, no prior work simultaneously (*i*) embeds both prompts and arbitrary experts into a unified metric space, (*ii*) incorporates inference cost explicitly into its learning objective, and (*iii*) generalizes effectively to new LLMs and out-of-distribution prompts using simple, efficiently computable descriptors.

# 3 Method

This section formalizes our *Cost-Spectrum Contrastive Router* (CSCR) and its two drop-in, model-agnostic descriptors: *logit fingerprints* and *perplexity fingerprints*. CSCR is trained once on a fixed pool of LLMs and deployed without modification on any subset of that pool. At inference time it performs a $k$-NN lookup in a FAISS [21] index [2] to return the $k$ most cost-effective experts for a prompt. Throughout this section, let $\mathcal{H} = \{h^{(1)}, \ldots, h^{(M)}\}$ denote the available LLMs, $c(h)$ their normalized cost, and $\Phi(\mathbf{x}) \in \mathbb{R}^D$ our frozen query encoder with a trainable MLP head $g_\theta(.)$.

## 3.1 Model Fingerprints

We map every LLM to a compact, task-independent vector $\mathbf{d}_h \in \mathbb{R}^{D'}$. Both descriptors are gradient-free. They can be computed off-line, cached, and shipped without IP-sensitive weights.

### 3.1.1 Logit-Footprint Descriptors for Transparent LLMs

Let $S_{\mathrm{probe}} = \{x^{(i)}\}_{i=1}^N$ be a fixed set of short, diverse prompts shared across all experts. For an autoregressive LLM $h$, denote by

$$p_h(v \mid x, t) = \mathrm{softmax}\big(\mathrm{logits}_h(x)_t\big)_v, \quad v \in \mathcal{V}, \ t \geq 1 \tag{1}$$

the probability that $h$ emits vocabulary token $v$ at generation step $t$ conditioned on the prompt prefix $x$. We compress these probabilities into a fixed-length logit footprint:

$$\mathbf{d}_{\mathrm{logit}}(h) = \frac{1}{NT} \sum_{i=1}^N \sum_{t=1}^T \big[p_h\big(v_k \mid x^{(i)}, t\big)\big]_{k=1}^K \in \mathbb{R}^K, \tag{2}$$

where $T$ is a small horizon, and $\{v_k\}_{k=1}^K$ are the $K$ most frequent tokens across all probes. We $\ell_2$-normalize $\mathbf{d}_{\mathrm{logit}}(h)$ to live on the unit hypersphere, after which cosine similarity is a proxy for KL divergence between the first token distributions.

---

[2]We use a FAISS `IndexFlatIP`. https://github.com/facebookresearch/faiss/wiki/Faiss-indexes

**Why logits?** Equation (2) directly samples the model's internal predictive distribution $p_h(\cdot)$—the very quantity trained by maximum-likelihood objective $-\sum \log p_\theta$ [6]. It therefore encodes both topical preference and generation style while remaining inexpensive (only $N \times T$ forward passes with greedy decoding). Thus, we use (2) as the primary descriptor whenever logits are available. See Appendix C.1 for a more detailed discussion.

### 3.1.2 Perplexity Fingerprints for Black-Box or API-Only LLMs

Closed-source APIs (e.g., GPT-o3 [37], Gemini-2.5 [69]) expose responses but hide almost all logits. For such models we adopt a per–prompt cross-entropy fingerprint:

$$\ell_h(x) = -\frac{1}{L_x} \sum_{j=1}^{L_x} \log p_h\big(w_j \mid w_{<j}\big), \tag{3}$$

$$\mathbf{d}_{\text{PPL}}(h) = \text{normalize}\Big(\big[\ell_h\big(x^{(i)}\big)\big]_{i=1}^N\Big) \in \mathbb{R}^N, \tag{4}$$

where $w_{1:L_x}$ are the gold target tokens (ground-truth answers if available, or probe continuations); $\text{normalize}(\cdot)$ denotes mean-centering and unit-variance scaling. Practically, we approximate (3) with a lightweight open LM that scores the API output $\hat{y}_h(x)$ instead of inaccessible $p_h$:

$$\tilde{\ell}_h(x) = -\frac{1}{|\hat{y}_h(x)|} \sum_j \log p_{\text{gpt2}}\big(\hat{y}_{h,j} \mid \hat{y}_{h,<j}\big). \tag{5}$$

See Appendix C.2 for a more detailed discussion.

**Why perplexity?** Cross-entropy is proportional to $\text{KL}(p_{\text{true}} \| p_h)$ plus entropy of the data distribution. It therefore quantifies text fit and has long been a proxy for LM quality [58, 66]. Prior work has shown that perplexity can serve as an effective metric for distinguishing between human-generated text and LLM-generated output [31]. Moreover, although $\tilde{\ell}_h$ is an approximation, the descriptor vector in (4) still captures how hard each prompt is for a given expert. So routing on these vectors recovers much of the benefit of logit footprints while remaining viable for black-box LLMs. Figure 2 shows that, despite using the same model to compute the perplexity of text generated by different LLMs, we still observe a clear separation between the expert descriptors.

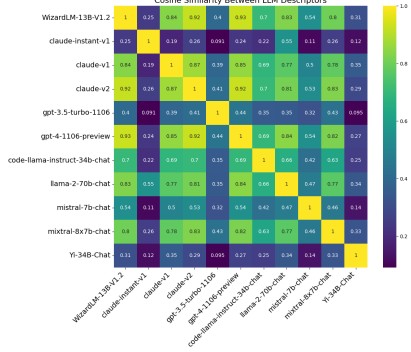

Figure 2: Cosine similarity of perplexity descriptors for RouterBench LLMs. Despite using a shared scorer, the descriptors distinctly separate the experts.

**Unified metric space.** We can make both descriptors reside on the unit sphere $\mathbb{S}^{K-1}$ by setting $N = K$ (i.e., the number of top tokens in logit fingerprints equals the number of prompts in perplexity fingerprints) with cosine similarity $\sigma(\mathbf{d}_1, \mathbf{d}_2) = \mathbf{d}_1^\top \mathbf{d}_2$. A single $k$-NN router can thus mix open-weight experts (logit fingerprints) and API experts (perplexity fingerprints) without altering downstream training loss and only the descriptor extraction pipeline has to change (See Table 4) .

### 3.2 Cost-Spectrum Contrastive Router

A contrastive router learns a shared embedding space where each query vector is pulled toward the descriptor of the right-sized expert and pushed away from less suitable ones. This yields three key pay-offs. First, because routing reduces to a nearest-neighbor lookup in that space, inference is a microsecond operation that adds virtually no latency to serving large expert pools. Second, contrastive objectives let the router exploit implicit supervision ("expert X solved this prompt while expert Y failed" or "X is cheaper than an equally accurate Y") so it can be trained with only correctness or cost signals with no dense human annotations required. Third, the geometry learned by contrastive

learning naturally generalizes: queries that look semantically or structurally similar land near the same expert regions, which improves robustness to distribution shift and unseen prompts, a behavior long noted in contrastive representation learning. Together, these properties make a contrastive router an efficient, supervision-light and highly adaptable choice for directing traffic in modern multi-LLM systems.

### 3.3 Background: The Classic INFONCE Loss.

Let a minibatch contain $B$ queries $\{\mathbf{x}_i\}_{i=1}^{B}$ and a memory bank of $M$ keys $\{\mathbf{e}_m\}_{m=1}^{M}$. A query encoder $f_\theta : \mathcal{X} \to \mathbb{R}^d$ produces representations $\mathbf{q}_i = f_\theta(\mathbf{x}_i)/\|f_\theta(\mathbf{x}_i)\|_2$, and the keys are $\ell_2$-normalized in advance, $\mathbf{e}_m = \mathbf{E}_m/\|\mathbf{E}_m\|_2$ (here, $E_m$ is the expert descriptor from Equation (2) or (4), i.e. $E_m = d(h_m)$). For each query $i$ let $\mathcal{P}(i) \subset [M]$ denote the positives (e.g. correct experts) and $\mathcal{N}(i) = [M] \setminus \mathcal{P}(i)$ the negatives. The vanilla InfoNCE objective [87] maximizes a log-softmax over cosine similarities

$$\mathcal{L}_{\text{InfoNCE}} = -\frac{1}{B}\sum_{i=1}^{B}\log\frac{\displaystyle\sum_{m\in\mathcal{P}(i)}\exp\!\left(\frac{\mathbf{q}_i^{\top}\mathbf{e}_m}{\tau}\right)}{\displaystyle\sum_{m'=1}^{M}\exp\!\left(\frac{\mathbf{q}_i^{\top}\mathbf{e}_{m'}}{\tau}\right)}, \tag{6}$$

where $\tau > 0$ is a temperature. It encourages queries to be close to any positive but far from all negatives, thereby learning a metric embedding.

### 3.4 Cost-Spectrum InfoNCE.

**Why Incorporate Cost:** Routing must balance two competing axes: *quality* (the expert is correct) and *inference cost* $c_m$ (e.g. dollars or latency). The classical InfoNCE loss ignores $c_m$, so the encoder can satisfy the objective by clustering any correct experts—most often the cheapest ones, since there are usually more of them in the pool—around the query embedding. Compounding this, easier prompts tend to occur more frequently, so training examples are skewed toward cases where cheap experts suffice. Once those low-cost positives are nearby, there is no training signal to learn where the slightly more expensive but markedly more accurate models live. Empirically, this drives the router to over-use the bargain-bin checkpoints, hurting accuracy and leaving significant potential untapped, even though paying a little more would buy a large quality jump (see Table 5).

We therefore introduce a **cost-aware spectrum** version that:

1. Selects all positive per cost band, preventing domination by extremely cheap or extremely costly experts

2. Assigns band-specific temperatures so that harder (costlier) positives yield smoother gradients

3. Penalizes negatives proportionally to their cost, pushing the encoder to prefer cheaper mistakes if it must err.

Formally we first normalize costs $c_m \in [0,1]$ and partition them into $K$ disjoint percentile bands $\mathcal{B}_k = \{m : c_m \in [\beta_k, \beta_{k+1})\}$ with quantiles $\beta_0 = 0 < \cdots < \beta_K = 1$. For each query $i$ and band $k$, let $\mathcal{P}_{ik} = \mathcal{P}(i) \cap \mathcal{B}_k$ denote the set of correct experts in that band. All experts in $\mathcal{P}_{ik}$ are treated as positives, and weighted by a softmax over similarities scaled by a band-specific temperature

$$\tau_k = \tau_{\min} + \alpha \cdot \bar{c}_k, \tag{7}$$

where $\bar{c}_k$ is the mean cost of experts in $\mathcal{B}_k$.

With $\Phi(\mathbf{x}) \in \mathbb{R}^D$ being our frozen query encoder with a lightweight trainable MLP head $g_\theta(.)$, the loss for a query $\mathbf{q}_i = g_\theta(\Phi(\mathbf{x}_i))/\|g_\theta(\Phi(\mathbf{x}_i))\|_2$ is then an average over all non-empty bands:

$$\ell_i^{\text{CS}} = -\frac{1}{|\mathcal{K}i|} \sum_{k \in \mathcal{K}i} \log \frac{\sum_{m \in \mathcal{P}_{ik}} \exp\left(\frac{\mathbf{q}_i^\top \mathbf{e}_m}{\tau_k}\right)}{\sum_{m'=1}^{M} \exp\left(\frac{\mathbf{q}_i^\top \mathbf{e}_{m'} - \gamma c_{m'}}{\tau_k}\right)}, \tag{8}$$

where $\mathcal{K}_i$ is the set of cost bands that contain at least one positive and $\gamma \geq 0$ controls the negative cost penalty. Averaging over the minibatch yields $\mathcal{L}_{\text{CS}} = \frac{1}{B} \sum_{i=1}^{B} \ell_i^{\text{CS}}$.

**Banded positives.** By retaining all positives within each cost band, we ensure that high-cost, correct experts still receive gradient signal, even when low-cost models also answer correctly. This prevents cost-collapse, the failure mode discussed in 3.4, where training signal concentrates on cheap experts due to prompt and model imbalances.

**Cost-dependent temperature.** Higher bands (larger $\bar{c}_k$) get larger $\tau_k$, flattening their softmax and avoiding vanishing gradients when few difficult positives exist. In contrast, cheap bands keep a low temperature, sharpening the push towards inexpensive correct experts.

**Negative cost penalty.** Subtracting $\gamma c_m$ in the denominator (not the numerator) means that expensive wrong experts contribute more to the partition function, hence increase the loss; the encoder is thus encouraged to separate from them first.

Thus we align three signals in the same metric space: *(i)* semantic proximity via the query encoder, *(ii)* expert capability via fingerprints, and *(iii)* user preference via cost scaling. Previous cost-aware objectives for retrieval weight the final scoring function at inference time (e.g. [48, 40]). Our formulation also integrates cost during representation learning, inducing a feature geometry that naturally interpolates accuracy and cost. Equation (8) collapses to standard InfoNCE when $K = 1$ and $\gamma = 0$. See Appendices C.3 and C.4 for a more detailed discussion.

### 3.5 Inference Router

Given a test prompt $\mathbf{x}$ we retrieve

$$\hat{r}(\mathbf{x}) = \arg\max_{h \in \text{Top}_k(\mathbf{x})} \left[\cos\langle g_\theta(\Phi(\mathbf{x})), \mathbf{d}_h\rangle - \lambda\, c(h)\right], \tag{9}$$

$\lambda$ is the cost weight and $\text{Top}_k(\mathbf{x})$ retrieves the $k$ most similar experts to the prompt from the FAISS index. We use $k = 4$ by default. During training, a similar composite score appears in the cost-spectrum InfoNCE objective (Equation (8)), allowing the encoder to rank candidate models by similarity to expert descriptors plus the cost term $\gamma c(h)$, just as in the inference rule of Equation 9. We show in the next section that this alignment between training and inference is highly effective.

## 4 Experiments

### 4.1 Experimental Settings

**Baselines** We compare our proposed method against a comprehensive set of baselines. Specifically, we include UMR [40], a recent technique that clusters prompt embeddings to route queries to LLM pools efficiently; Thompson Sampling [48, 2], which frames routing as a bandit exploration–exploitation problem to balance cost and accuracy dynamically; Pareto-optimal routing [34], a strategy that selects models by explicitly considering the cost-accuracy Pareto frontier; and two extreme baselines: Random, which selects models uniformly at random to represent naive routing without intelligent selection, and Oracle [40], which always selects the most accurate model at the lowest possible cost and thus represents a theoretical performance ceiling. Additionally, we evaluate against parametric-softmax gating methods inspired by mixture-of-experts architectures(e.g. [62]) and SoftMoE [64], which models router decisions via differentiable soft gating functions.

**Datasets & Benchmarks** We train our router and evaluate it on three datasets: EmbedLLM [105], MixInstruct [39], and RouterBench [34]. For EmbedLLM and MixInstruct, we sample 192 probes from their respective validation sets. Each probe is processed to extract logit-based descriptors by

| Router | EmbedLLM | | | Mix-Instruct | | | RouterBench | | |
|---|---|---|---|---|---|---|---|---|---|
| | AUDC ↑ | QNC ↓ | Peak ↑ | AUDC ↑ | QNC ↓ | Peak ↑ | AUDC ↑ | QNC ↓ | Peak ↑ |
| Oracle (upper bound) | 0.960 | 2.87 | 0.979 | 0.079 | 10.17 | 0.081 | 0.891 | 0.290 | 0.910 |
| UMR [40] | 0.515 | 58.61 | 0.562 | 0.049 | 10.35 | 0.050 | 0.568 | 0.487 | 0.615 |
| Thompson [2] | 0.472 | 48.81 | 0.514 | 0.044 | 10.09 | 0.045 | 0.622 | 1.634 | 0.787 |
| Soft-MoE [64] | 0.404 | 70.00 | 0.577 | 0.030 | 10.09 | 0.045 | 0.599 | 1.659 | **0.794** |
| Parametric | 0.506 | 70.00 | 0.555 | 0.039 | 12.00 | **0.052** | 0.691 | 1.658 | **0.794** |
| Pareto-Random [34] | 0.369 | 16.03 | 0.393 | 0.036 | 6.16 | 0.043 | 0.5172 | 0.589 | 0.545 |
| Random | 0.379 | 18.09 | 0.392 | 0.037 | 6.16 | 0.043 | 0.5147 | 0.585 | 0.542 |
| **CSCR (Ours)** | **0.541** | 43.28 | **0.578** | **0.051** | 9.32 | **0.052** | **0.7110** | 1.660 | **0.794** |

Table 1: Deferral curve metrics across three benchmarks. Our Cost-Spectrum Contrastive Router achieves the highest area under the deferral curve (AUDC), competitive or superior peak accuracy and lower quality-neutral cost (QNC) compared to key baselines. The Oracle router serves as an upper bound, retrospectively selecting the lowest-cost LLM that yields the correct answer.

capturing the top $K = 256$ tokens over a horizon of $T = 10$ tokens (Equation (2)). For RouterBench, we sample 192 probes from its training set. We compute perplexity-based descriptors on RouterBench and use GPT-2 [66]. On both EmbedLLM and RouterBench, we use binary accuracy as the per-sample evaluation metric. For MixInstruct, we employ exponentiated BARTScore [97] as the evaluation metric, following the approach in prior work [40, 39].

**Training**  We use a frozen `sentence-transformers/all-MiniLM-L6-v2`[70] model as the embedding backbone across all experiments. Our trainable router component is a two-layer MLP which projects prompt embeddings into the expert descriptor space. We train our contrastive router on the training splits of each dataset. For the cost spectrum loss (Equation (8)), we set the number of cost bands $K = 5$ and the negative cost penalty $\gamma = 0.2$. The hyperparameters for band-specific temperatures (Equation (7)) are set as $\alpha = 0.25$ and $\tau_{\min} = 0.05$. See D.1 for full details.

**Evaluation**  We evaluate each routing strategy using a deferral curve [40] which plots the average response quality against the total inference cost. Sweeping the routing penalty parameter $\lambda$ over the interval $\lambda \in [0, \lambda_{\max}]$ (Equation (9)) traces the deferral curve. For the EmbedLLM and MixInstruct datasets, we define the cost of processing a prompt as the number of parameters in the LLM, a proxy for computational resources and latency. In the case of RouterBench, we utilize the actual API call costs in USD, as provided in the dataset. Following [40] we employ evaluation metrics including Area Under the Deferral Curve (AUDC), peak accuracy and Query-Normalized Cost (QNC), the minimum relative cost required to match the performance of the most accurate tested LLM.

## 4.2  Results

Table 1 presents Deferral curve metrics across the benchmarks. Our CSCR consistently outperforms all relevant baselines, achieving the highest AUDC and demonstrating competitive or superior peak accuracy. Notably, it attains lower QNC, indicating more cost-effective routing decisions. These results show the effectiveness of our cost-aware router learning approach in balancing performance and inference cost. The Oracle router, which selects the optimal LLM for each query, establishes an upper bound for performance. The benchmarks are dominated by lower cost experts, hence the lower QNC for random baselines. See Appendix D.6 for results on statistical significance.

### 4.2.1  Generalization to New LLMs

We also evaluate the robustness of our method and baselines in scenarios where new LLMs are introduced during testing. Specifically, we select two-thirds of the EmbedLLM training models to train our router and the baselines following [40], using only responses from these selected models. Table 2 summarizes the performance metrics, while Figure 3 illustrates the corresponding deferral curves. Testing is conducted exclusively on the unseen LLM pool. Our results indicate that our approach exhibits superior robustness under these conditions.

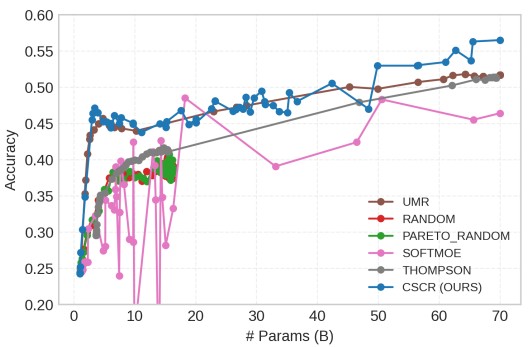

Figure 3: Deferral curves of test on new LLMs.

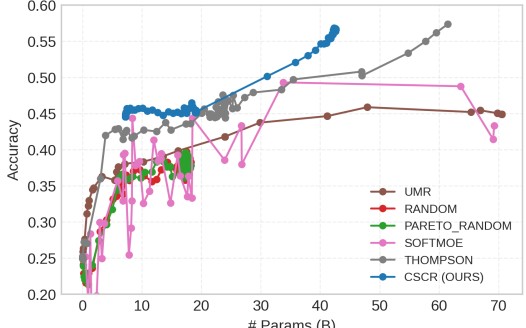

Figure 4: Deferral curve of OOD prompts.

| Router | EmbedLLM | | |
| --- | --- | --- | --- |
| | AUDC ↑ | QNC ↓ | Peak ↑ |
| Oracle (upper bound) | 0.9111 | 4.118 | 0.951 |
| UMR | 0.4766 | **64.300** | 0.518 |
| Thompson | 0.4478 | 68.904 | 0.514 |
| Soft-MoE | 0.4109 | 18.282 | 0.485 |
| Pareto-Random | 0.3812 | 15.662 | 0.407 |
| Random | 0.3829 | 15.372 | 0.403 |
| **CSCR (Ours)** | **0.4848** | 70.000 | **0.565** |

Table 2: Deferral curve metrics on new LLMs. CSCR shows better robustness to new LLMs.

| Router | EmbedLLM | | |
| --- | --- | --- | --- |
| | AUDC ↑ | QNC ↓ | Peak ↑ |
| Oracle (upper bound) | 0.9511 | 2.872 | 0.979 |
| UMR | 0.4249 | 47.910 | 0.459 |
| Thompson | 0.4863 | 69.983 | **0.574** |
| Soft-MoE | 0.4207 | 70.017 | 0.555 |
| Pareto-Random | 0.3676 | 17.417 | 0.396 |
| Random | 0.3717 | 17.499 | 0.397 |
| **CSCR (Ours)** | **0.5146** | **42.338** | 0.568 |

Table 3: Deferral curve metrics on OOD prompts. CSCR shows superior robustness.

### 4.2.2 Generalization to Out-of-Distribution Prompts

We evaluate our approach on an *out-of-distribution (OOD) prompt at test time* scenario. We split the EmbedLLM dataset into two subsets: one focusing on STEM-related prompts (e.g., science and technology) and the other comprising all remaining categories (see Appendix D.2.1 for detailed experimental settings). As illustrated in Figure 4 and quantified in Table 3, CSCR significantly outperforms all baselines across all key metrics. Specifically, our router achieves an AUDC of 0.5146 compared to the next-best baseline, Thompson, at 0.4863, highlighting its superior robustness and accuracy when handling diverse OOD prompts. This performance advantage demonstrates that our method generalizes exceptionally well, maintaining reliable decision-making capability across varied and harsh distributional shifts.

### 4.3 Ablative Studies

### 4.3.1 Descriptor Choice

Table 4 compares two descriptors on Mix-Instruct, the only benchmark where we can compute both. Perplexity descriptors (obtained by running every candidate answer through an auxiliary language model) lift the AUDC from 0.461 to 0.467. The absolute peak accuracy, however, changes less than 0.1%. Because the perplexity pipeline requires (i) generating text with the model in the pool and (ii) a second forward pass through a public LM, it is often at least 2× slower. Logit descriptors, in contrast, need only a single pass on open-weights models and still deliver competitive AUDC. We therefore adopt logit-based descriptors for all open LLMs and fall back to perplexity descriptors only when logits are inaccessible. The "mixed" row in 4 represents the results obtained by using logit descriptors for 6 randomly selected LLMs and perplexity descriptors for the remaining 5. This shows that both descriptors can be combined within the same pool without negatively affecting the results. In fact, performance slightly improves compared to using only a single descriptor type. This observation further supports our discussion of the "unified metric" in Section 3.1.2.

### 4.3.2 Cost-Aware Training

| Router | MixInstruct | | |
|---|---|---|---|
| | AUDC ↑ | QNC ↓ | Peak ↑ |
| Logit Desc. | 0.0461 | 6.426 | 0.046 |
| Perp. Desc. | 0.0467 | 8.967 | 0.047 |
| Mixed | 0.0473 | 6.233 | 0.047 |

Table 4: Effect of Descriptor Type. Perplexity descriptors slightly improve AUDC but require an extra pass compared to the faster logit descriptors. Mixing descriptors has no impact on results.

| Router | EmbedLLM | | |
|---|---|---|---|
| | AUDC ↑ | QNC ↓ | Peak ↑ |
| Vanilla | 0.3421 | 6.382 | 0.362 |
| Cost-Aware | 0.4951 | 11.065 | 0.540 |

Table 5: Effect of cost-aware training: injecting cost awareness into the contrastive loss prevents the router from concentrating on cheap experts, and boosts the AUDC and peak accuracy of the router. The trade-off is a higher QNC.

Replacing the vanilla InfoNCE loss with our cost-spectrum variant significantly increases the AUDC (0.342 → 0.495) and raises the peak attainable accuracy from 36% to 54% as shown in Table 5. The trade-off is a higher Quality-Neutral Cost QNC, meaning the router now leans more on expensive but accurate models; however, the large AUDC gain shows that, for any realistic cost budget, users receive better accuracy-per-dollar overall. This confirms our discussion in Section 3.4, that explicitly teaching the encoder to respect the cost hierarchy of experts is crucial.

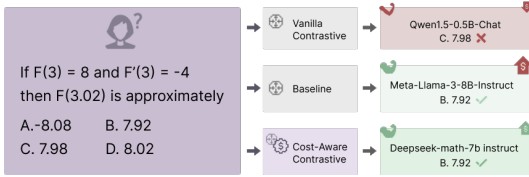

Figure 5: Real sample of routing decisions made by UMR, vanilla contrastive router, and CSCR. CSCR chooses a more expensive but accurate expert than the vanilla router, while also selecting a cheaper option than UMR, achieving better accuracy–cost trade-offs on both ends.

Furthermore, Figure 1 and Table 1 show that while cost-aware contrastive training incurs more cost than the vanilla variant, it remains far more efficient than the baselines, achieving lower QNC by learning to distinguish good cheap experts from both bad cheap and bad expensive ones. See Figure 5 for an example.

### 4.3.3   Cost-Spectrum Granularity

Figure 6 shows that performance varies with the number of cost bands. Using 5 bands yields the best results, with the highest AUDC and Peak accuracy, suggesting a good balance between flexibility and generalization. Fewer bands (e.g., 2) limit routing precision, while too many bands (e.g., 15 or 20) degrade performance, likely due to over-fragmentation and increased decision noise. This highlights the importance of tuning the number of bands per dataset to avoid both under- and overfitting. See Appendix D.2 for more ablations and detailed discussion.

Figure 6: Deferral curves across different numbers of cost bands.

## 5   Theoretical Analysis

**Set-up.** Let $\mathcal{X}$ be the space of prompts and $\mathcal{H} = \{h_1, \ldots, h_M\}$ a fixed pool of experts with per-call cost $c(h) \in \mathbb{R}_{>0}$. For a query $x \in \mathcal{X}$ and ground-truth $y$ we write $\ell(x, y, h) = \mathbb{1}[h(x) \neq y]$ for the 0-1 loss and $\gamma(x, h) = \mathbb{P}_{y|x}[\ell(x, y, h) = 1]$ for the Bayes error. The *cost-adjusted risk* of a router $r$ is

$$\mathcal{R}_\lambda(r) = \mathbb{E}_{(x,y)\sim\mathcal{D}}\Big[\ell\big(x, y, h_{r(x)}\big) + \lambda\, c\big(h_{r(x)}\big)\Big],$$

where $\lambda \in \mathbb{R}_{\geq 0}$ trades accuracy for cost [23].

Our router embeds *queries* via $\Phi_q : \mathcal{X} \to \mathbb{R}^d$ and *experts* via $E = [e_1; \ldots; e_M] \in \mathbb{R}^{M \times d}$, using either (i) logits fingerprints (EmbedLLM, Mix-Instruct) or (ii) perplexity fingerprints (RouterBench). Given a query $x$, the $k$-NN rule selects

$$\hat{r}_k(x; \lambda) = \underset{m \in [M]}{\arg\min} \Big[ \underbrace{\tfrac{1}{k} \sum_{j \in \mathcal{N}_k(x)} \gamma(x_j, h_m)}_{\text{local error}} + \lambda\, c(h_m) \Big].$$

where $\mathcal{N}_k(x)$ are the $k$ nearest training prompts to $x$ in $\|\Phi_q(\cdot)\|_2$.

## 5.1 Excess-risk of cost-spectrum k-NN

**Assumption 5.1** (Lipschitz Bayes error). *There exists $L > 0$ s.t. for all $x, x' \in \mathcal{X}$ and $h \in \mathcal{H}$, $|\gamma(x, h) - \gamma(x', h)| \le L \|\Phi_q(x) - \Phi_q(x')\|_2$.*

**Theorem 5.2** (Excess risk). *Let $\hat{r}_k$ be trained on $n$ i.i.d. prompt embeddings. Under Assumption 5.1, for any $\lambda \ge 0$ and any $k \le n$,*

$$\mathbb{E}\big[\mathcal{R}_\lambda(\hat{r}_k)\big] - \mathcal{R}_\lambda(r^\star) \ \le \ C\Big( \sqrt{\tfrac{k}{n}} + k^{-1/d} \Big),$$

*where $C$ depends only on $L$ and the diameter of $\Phi_q(\mathcal{X})$, and $r^\star$ is the Bayes-optimal rule $r^\star(x) = \arg\min_m \big[\gamma(x, h_m) + \lambda c(h_m)\big]$.*

The proof follows the classical $k$-NN bound of [53, 5] with an extra $\lambda c(h)$ term that is constant w.r.t. $x$ and therefore preserves the rate.

## 5.2 Consistency of Cost-Spectrum InfoNCE

Fingerprints live on $\mathbb{S}^{d-1}$, so dot products are scaled similarities $S_{im} = \frac{q_i^\top e_m}{\tau_k}$ with *band-dependent temperature* $\tau_k$ (Equation (7)). Let $\beta_0 = 0 < \beta_1 < \cdots < \beta_K = 1$ partition costs into bands $\mathcal{B}_k = \{m : c(h_m) \in [\beta_k, \beta_{k+1})\}$, and denote $\mathcal{P}_{ik} = \mathcal{P}(i) \cap \mathcal{B}_k$ the correct experts for query $i$ that fall in band $k$. The *Cost-Spectrum* InfoNCE loss for a single query $i$ is

$$\ell_i^{\mathrm{CS}} = -\frac{1}{|\mathcal{K}_i|} \sum_{k \in \mathcal{K}_i} \log \frac{\displaystyle\sum_{m \in \mathcal{P}_{ik}} \exp\big(S_{im}\big)}{\displaystyle\sum_{m'=1}^{M} \exp\big(S_{im'} - \gamma\, c_{m'}\big)},$$

where $\mathcal{K}_i = \{k : \mathcal{P}_{ik} \ne \varnothing\}$ and $\gamma \ge 0$ is the negative cost penalty.

**Lemma 5.3** (Directional alignment with cost bands). *At any stationary point of $\mathcal{L}_{\mathrm{CS}} = \frac{1}{B} \sum_i \ell_i^{\mathrm{CS}}$, for every query $i$ and any $m^+ \in \mathcal{P}_{ik}$, $m^- \in \mathcal{N}(i)$ with $c_{m^+} \le c_{m^-}$, $q_i^\top e_{m^+} > q_i^\top e_{m^-}$.*

Lemma 5.3 shows the optimum ranks cheaper correct experts ahead of expensive or wrong ones, explaining the empirical benefit of the cost term.

## 5.3 Discussion

Theorem 5.2 guarantees that *if* query–expert descriptors are Lipschitz, CSCR converges to the Bayes-optimal router at the usual $k$-NN rate. Lemma 5.3 justifies the specific form of our InfoNCE objective. Sec. 4 verifies these claims on three benchmarks. See proofs in Appendix B.

# 6 Conclusion

We presented CSCR, a simple and efficient framework for cost-aware routing across a pool of LLMs. Our method uses two lightweight expert descriptors and trains a contrastive encoder to select the cheapest accurate expert within adaptive cost bands. Despite its simplicity, CSCR outperforms more complex routing baselines and demonstrates strong generalization to unseen models and out-of-distribution prompts. Our findings highlight the importance of embedding cost-awareness directly into the training objective of routers, rather than deferring it to test time. As model pools grow in size and diversity, activating the right-sized expert per query is critical for minimizing latency and cost. We see CSCR as a step toward more sustainable and adaptive LLM deployments, and believe this line of research is essential to avoid defaulting to unnecessarily large models for simple tasks.

# 7 Acknowledgments

The authors would like to thank Zahra Miri for her assistance in preparing the figures.
This work was made possible by NSF IIS 2347592, 2348169, DBI 2405416, CCF 2348306, CNS 2347617.

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

## Limitations and Broader Impact

This paper proposes a routing framework that improves inference efficiency in large language models (LLMs). By directing simple queries to smaller models, it cuts computation and memory overhead, lowering costs and environmental impact. Although the framework is built for large-scale deployments, we could not test very large LLM pools due to resource limits. Still, our experiments validate the concept. Future work will examine routing across specialized subnetworks and conditional computation within a single LLM. Moreover, future work can answer: How big must a model be to recognize a problem's difficulty even if it can't solve the problem itself? Future work can study whether size and fine-tuning helps, would RL on LLMs makes them able to purely as routers, and whether there is a model size when difficulty awareness kicks in.

## A  Related Work

### A.1  Enhancing and Optimizing LLMs

Large language models (LLMs) have demonstrated remarkable capabilities across diverse NLP tasks [66, 6]. To further improve their performance and efficiency, numerous strategies have been proposed.

**Single-LLM Techniques.**  Enhancement approaches targeting individual LLMs include fine-tuning [67], prompting strategies like Chain-of-Thought (CoT) [102, 90], Tree-of-Thoughts [96], and inference-acceleration techniques such as early exiting [84, 103, 72] and speculative decoding [77, 83, 9, 45, 8]. Additionally, Mixture-of-Experts (MoE) architectures [36, 41, 75, 24, 104, 38] route inputs through sparse sub-models or "experts", reducing cost while retaining performance. However, these methods typically operate within a single LLM's structure and may not generalize to multi-model scenarios.

**Model Fusion and Merging.**  Fusion strategies synthesize outputs from multiple LLMs to improve output quality [68, 39, 29, 89, 52]. Fusion approaches often rely on unsupervised metrics [99, 73, 97] or ensemble voting to determine the final output [44]. A related but distinct technique is model merging [50], where weights from multiple pre-trained or fine-tuned models are combined—either directly via methods like weight averaging [92], Task Arithmetic [35], or Fisher merging [56]; or using more sophisticated techniques like TIES [94], AdaMerging [95], and ZipIt [78].

**Cascading.**  In contrast, cascading invokes models sequentially—often ordered by computational cost—and halts once a satisfactory output is generated [10, 98, 10, 30]. Such approaches strike a balance between quality and efficiency, making them particularly attractive in production settings.

### A.2  LLM Routing

Routing methods dynamically select the most appropriate LLM from a pool for each input, aiming to optimize performance and cost without querying all models. Two primary strategies dominate: non-predictive and predictive routing.

**Non-Predictive Routing.**  Non-predictive methods generate outputs from one or more models before making a selection. FrugalGPT [10] exemplifies this category, using a sequential strategy and a response quality threshold to minimize cost. Other works adopt layered inference architectures to escalate hard queries to more powerful models [91], or leverage cascades with self-verification [54, 98, 43, 51].

**Predictive Routing.**  In contrast, predictive routing aims to select the best model *before* any inference is performed. Strategies include supervised learning [76], reward-model-based routing [32, 51], and meta-models trained to predict LLM performance given an input [71]. Router models vary widely in implementation, including neural networks [19, 88, 11, 1], $k$-nearest neighbors [34, 76, 80, 43], matrix factorization [62, 105, 48], and graph neural networks [25]. Others incorporate model-specific tokens or train across multiple domains [18, 7].

**Theoretical Foundations and Robustness.** Routing and cascading are grounded in broader litera­ture, including selective classification [27, 60], learning to defer [55], and learning to reject [12, 4, 14]. Several works explore supervision levels [51, 101], robustness [16, 59, 74], and evaluation frame­works for routers [33, 32].

### A.3 Routing as Recommendation

Routing can also be framed as a recommendation problem, wherein the input query plays the role of a "user", the pool of LLMs corresponds to "items", and past performance metrics form the implicit interaction history [100, 93]. However, unlike conventional recommender systems, routing has limited "user" features (i.e., input metadata), making label collection and generalization especially challenging [61, 49].

Matrix factorization, attention-based models, and graph neural networks are used in both recom­menders and routers [62, 105, 25], reinforcing the close link between the two domains.

### A.4 Scaling Laws and Architecture Trends

Scaling laws [42, 47] describe predictable trends between model size, data, and performance, guiding the development of efficient LLM architectures. These insights have been extended to MoEs [24, 13], sparse models [26], and hybrid systems [28, 63], offering context for when routing or merging approaches might be most beneficial.

### A.5 Routing within MoE and Hybrid Architectures

Routing is a central mechanism in MoE models [36, 41, 75], where expert modules are dynami­cally activated based on input. While classical MoEs involved equally-sized sub-models, modern approaches like Switch Transformer [24] and Mixtral [38] employ sparse activation to minimize cost.

Routing LLMs can be viewed as a coarse-grained MoE, where each expert is a full LLM. Approaches like UltraFuser [20], Branch-Train-MiX [81], and token-level fusion highlight recent advances in combining model specialization and flexibility.

## B   Proofs of Theoretical Results

### B.1   Notation and Preliminaries

Let the Bayes-optimal router be $r^\star(x) = \arg\min_{m \in [M]} \big[\gamma(x, h_m) + \lambda c_m\big]$ and define the *excess cost–error gap*

$$\Delta_m(x) = \gamma(x, h_m) + \lambda c_m - \big(\gamma(x, h_{r^\star(x)}) + \lambda c_{r^\star(x)}\big).$$

Hence $\Delta_{r^\star(x)}(x) = 0$ and $\Delta_m(x) \geq 0$. For a query $x$ let $r_k(x)$ be the radius of the ball $B(x, r) \subset \mathbb{S}^{d-1}$ (in the cosine metric) that contains the $k$-th nearest training neighbor. If the marginal on $q(\mathcal{X})$ has a density, $\mathbb{E}\big[r_k(x)^d\big] \leq C_1 k/n$ [15, 79].

### B.2   Proof of Theorem 5.2

Writing $q_i = q(x_i)$ to lighten notation, decompose

$$\mathbb{E}\big[\mathcal{R}_\lambda(\hat{r}_k)\big] - \mathcal{R}_\lambda(r^\star) = \mathbb{E}_x \Bigg[\underbrace{\Delta_{\hat{r}_k(x)}(x) - \Delta_{\hat{r}_k(x)}(x_{j \in \mathcal{N}_k(x)})}_{(A)} + \underbrace{\Delta_{\hat{r}_k(x)}(x_{j \in \mathcal{N}_k(x)}) - \Delta_{r^\star(x)}(x_{j \in \mathcal{N}_k(x)})}_{(B)}\Bigg].$$

**Term (A): Lipschitz bias.** Assumption 5.1 gives $|\gamma(x, h) - \gamma(x', h)| \leq L\|q(x) - q(x')\|_2$, so $|(A)| \leq L\, r_k(x)$. Taking expectations and using $\mathbb{E}[r_k(x)] \leq (C_1 k/n)^{1/d}$ yields $\mathbb{E}[(A)] \leq C_2 k^{-1/d}$.

**Term (B): Estimation variance.** Let the empirical cost-adjusted risk be

$$\widehat{\Delta}_m(x) = \frac{1}{k} \sum_{j \in \mathcal{N}_k(x)} \big[\gamma(x_j, h_m) + \lambda c_m\big].$$

Hoeffding's inequality bounds

$$P\big(|\widehat{\Delta}_m(x) - \mathbb{E}[\widehat{\Delta}_m(x) \mid x]| \geq t\big) \leq 2e^{-2kt^2},$$

and a union bound over $m \leq M$ plus integration gives

$$\mathbb{E}[\max_m |\widehat{\Delta}_m(x) - \mathbb{E}[\widehat{\Delta}_m(x) \mid x]|] \leq C_3\sqrt{\tfrac{\log M}{k}}.$$

Because $\hat{r}_k(x)$ minimizes $\widehat{\Delta}_m(x)$, $(B) \leq 2\max_m |\widehat{\Delta}_m(x) - \mathbb{E}[\widehat{\Delta}_m(x) \mid x]|$. Combining with (A) implies

$$\mathbb{E}[\mathcal{R}_\lambda(\hat{r}_k)] - \mathcal{R}_\lambda(r^\star) \leq C\big(\sqrt{k/n} + k^{-1/d}\big).$$

## B.3   Proof of Lemma 5.3

For convenience write $S_{im} = q_i^\top e_m / \tau_k$ when $m \in \mathcal{B}_k$. The single-query loss (Equation (8)) is

$$\ell_i^{\mathrm{CS}} = -\frac{1}{|\mathcal{K}_i|} \sum_{k \in \mathcal{K}_i} \log \frac{\sum_{m \in \mathcal{P}_{ik}} \exp(S_{im})}{\sum_{m'=1}^M \exp\big(S_{im'} - \gamma c_{m'}\big)}.$$

Let

$$p_{im} = \exp(S_{im}) / \sum_{j \in \mathcal{P}_{ik}} \exp(S_{ij})$$

when $m \in \mathcal{P}_{ik}$ and

$$q_{im} = \exp(S_{im} - \gamma c_m) / \sum_{j=1}^M \exp(S_{ij} - \gamma c_j)$$

Then

$$\ell_i^{\mathrm{CS}} = -\tfrac{1}{|\mathcal{K}_i|} \sum_k \log \sum_{m \in \mathcal{P}_{ik}} p_{im}/q_{im}$$

Taking the gradient w.r.t. $q_i$ and setting it to zero gives $\sum_m (p_{im} - q_{im})e_m = 0$. Project onto $q_i$: $\sum_m (p_{im} - q_{im})S_{im} = 0$. Fix $m^+ \in \mathcal{P}_{ik}$, $m^- \in \mathcal{N}(i)$ with $c_{m^+} \leq c_{m^-}$. Because $p_{im^-} = 0$ while $p_{im^+} > 0$, the equality forces $q_{im^+} > q_{im^-}$, hence $S_{im^+} - \gamma c_{m^+} > S_{im^-} - \gamma c_{m^-}$. Rearranging yields $q_i^\top e_{m^+} > q_i^\top e_{m^-}$, establishing directional alignment.

$\square$

# C   Method

## C.1   Logit-Footprint Descriptors

**Why take the most frequent tokens?**   We use the most frequent tokens so the basis is shared and stable: they appear in all models, give low-noise estimates with few probes, and make calibration comparable across experts. They give less noisy estimates because they get non-negligible probability across many contexts, so their averaged log-probs vary less than rare or Out-of-Vocabulary tokens. In all experiments, we set $K = 256$ and $T = 10$ (D.1), which is large enough that the basis isn't dominated by a few function words.

**Are frequent tokens trivial?**   These tokens aren't used for their meaning. They're probes of each model's output behavior. Even common words get scored differently across models (temperature, punctuation/number handling, style). By averaging over many prompts and steps, the descriptor captures overall model behavior, not any single word's semantics. Also, a recent work [82] shows that LLMs exhibit stable, word-level idiosyncrasies (as the authors call them) that enable near-perfect model attribution using only the first few generated tokens (even after paraphrasing or translation), implying that common tokens still provide discriminative signals about a model's predictive calibration.

**Shared token set vs. per-model selection.** A shared basis gives all descriptors a common coordinate system. If each model used a different token set, cosine distances would mix basis changes with true behavior, hurting comparability. It would also require computing many more probes to align per-model bases that are different across models.

**Beyond raw frequency.** We deliberately kept the descriptors simple to isolate and quantify the contrastive router's contribution. Nonetheless, frequency is a pragmatic, not necessarily optimal, choice. Two variants that we considered and could be explored are:

- TF–IDF-weighted selection over the probe corpus.
- Picking tokens with the largest across-model log-prob variance.

These can be dropped into Equation 2 without changing downstream training or inference.

## C.2 The Step from Equation 3 to Equation 5

Equation (3) defines a per-prompt token NLL that requires access to an expert's next-token distribution $p_h(\cdot \mid \cdot)$. For API-only (black-box) experts, logits/probabilities are not exposed, so Equation (3) is not computable. Our remedy is to (a) let the API expert $h$ produce a deterministic continuation $\hat{y}_h(x)$ for prompt $x$ (greedy decoding), and (b) evaluate that sequence under a single shared, public scorer $p_S$ (kept fixed across all experts). This yields Equation (5), a *pseudo-perplexity*:

$$\tilde{\ell}_h(x) = -\frac{1}{|\hat{y}_h(x)|} \sum_{t=1}^{|\hat{y}_h(x)|} \log p_S(\hat{y}_{h,t} \mid \hat{y}_{h,<t}),$$

which we then normalize (similar to Equation (4)) and use as the fingerprint coordinate(s) for black-box experts.

If $\hat{y}_h$ is a typical (high-probability) output of $h$ (i.e., $\hat{y}_h \sim p_h$) then averaging the pseudo-perplexity $\tilde{\ell}_h(x) = -\frac{1}{|\hat{y}_h(x)|} \sum_t \log p_S(\hat{y}_{h,t} \mid \hat{y}_{h,<t})$ over many prompts/tokens is equivalent to taking an expectation over $y \sim p_h$:

$$\mathbb{E}_{y \sim p_h}\big[ -\log p_S(y) \big] = H(p_h, p_S) = H(p_h) + \mathrm{KL}\big(p_h \parallel p_S\big).$$

Here $H(p_h, p_S)$ is the cross-entropy of $p_h$ with respect to $p_S$, which decomposes into the entropy of $h$'s own distribution $H(p_h)$ and its divergence from the scorer $\mathrm{KL}(p_h \| p_S)$.

Because $p_S$ is fixed for all experts, $H(p_h, p_S)$ is a stable, model-specific quantity that makes descriptors comparable across experts ("same yardstick"). It is not the true NLL under $p_h$, but it preserves differences between experts via $H(p_h)$ and their mismatch to $p_S$ via $\mathrm{KL}(p_h \| p_S)$. In practice we use deterministic (greedy) decoding to reduce variance, averaging over many prompts/tokens makes the empirical $\tilde{\ell}_h(x)$ closely track the expectation above. Figure 2 empirically validates this argument.

## C.3 Band-Specific Temperatures and Smoother Gradients

Most prompts in everyday interactions (and in our datasets) can be handled by cheaper models; plus there are usually fewer very expensive experts overall. These expensive experts are only needed for a small fraction of hard prompts, so within those high-cost bands there are fewer suitable positives per query. With few positives, the similarity distribution becomes very peaky.

A larger $\tau_k$ in Equation 7 flattens the per-band softmax, reducing gradient variance and preventing the update from collapsing onto a single rare positive. Formally, for band $k$ the per-query gradient w.r.t. the query embedding is

$$\nabla_q \mathcal{L}_k = -\sum_{m \in P_{ik}} p_m^{(+)} \frac{e_m}{\tau_k} + \sum_{m'=1}^{M} p_{m'}^{(-)} \frac{e_{m'}}{\tau_k}, \tag{10}$$

where $p^{(+)}$ and $p^{(-)}$ are the band-restricted softmaxes over positives and all experts (with the negative cost penalty in the denominator). As $\tau_k$ increases, both softmaxes become less concentrated, so (i)

the gradient magnitude scales like $1/\tau_k$ and (ii) its direction is averaged over more positives, lowering variance across minibatches.

This is the sense in which band-specific temperatures yield smoother gradients. It is especially helpful in high-cost bands that otherwise have few positives and highly variable similarities. Without band-specific temperatures, the router can exhibit oscillatory updates on hard prompts (rare positives dominate, then vanish), slowing convergence and encouraging over-use of cheap experts. Empirically, we observed that bands and band-specific temperatures are important (Table 7)

### C.4 Dense Human Annotations

We considered using human annotations but intentionally avoided them: model pools change quickly, so adding/replacing experts would require fresh labels that are costly and often unavailable. Instead, we train with sparse correctness and cost signals, which remain portable across experts. If dense feedback is available, it could help in several ways:

- **Positive sets $P(i)$ with preference structure.** Replace binary "correct expert" labels with pairwise preferences (cheap-and-good $\succ$ expensive-and-similar $\succ$ clearly wrong), yielding band-aware positives and margin constraints. This can be implemented by expanding $P(i)$ and adding a lightweight pairwise ranking (DPO-style) regularizer within each cost band.

- **Difficulty-aware reweighting.** Use human "hardness" scores to upweight rare/hard prompts when computing the contrastive loss, especially in higher cost bands. this could balance the effective sample sizes across easy vs. hard (and cheap vs. expensive band) cases so the gradient isn't dominated by the abundant, easy examples.

- **Band calibration.** We can ask users how much quality they're willing to trade for a lower cost, then use that to set the cost bands and the penalty for picking more expensive models, so the router's choices match what users actually prefer.

## D Experiments

### D.1 Experimental Settings

**Baselines** We compare our proposed method against a comprehensive set of baselines designed to capture key routing strategies and their trade-offs. Specifically, we include UMR [40], a recent state-of-the-art technique that clusters prompt embeddings to route queries to LLM pools efficiently; Thompson Sampling [48, 2], which frames routing as a bandit exploration–exploitation problem to balance cost and accuracy dynamically; Pareto-optimal routing [34], a strategy that selects models by explicitly considering the cost–accuracy Pareto frontier; and two extreme baselines—Random, which selects models uniformly at random to represent naive routing without intelligent selection, and Oracle (Clairvoyant Upper-Bound [40]), which always selects the most accurate model at the lowest possible cost and thus represents a theoretical performance ceiling. Additionally, we evaluate against parametric gating methods (Parametric Softmax Router) inspired by classical mixture-of-experts architectures [62] and SoftMoE, which models router decisions via differentiable soft gating functions [64]. Collectively, these baselines enable us to rigorously assess whether our contrastive routing approach delivers meaningful improvements in performance, cost-efficiency, and generalization capabilities relative to existing strategies.

**Datasets, Benchmarks, and Evaluation** We train our router and evaluate it on three datasets: EmbedLLM [105], MixInstruct [39], and RouterBench [34]. For EmbedLLM and MixInstruct, we sample 192 probes from their respective validation sets. Each probe is processed to extract logit-based descriptors by capturing the top $K = 256$ tokens over a horizon of $T = 10$ tokens (Equation (2)), resulting in a 256-dimensional vector per model. For RouterBench, we sample 192 probes from its training set, ensuring these probes are excluded from the training data used for the contrastive router. We compute perplexity-based descriptors on RouterBench and use GPT-2 [66]. On both EmbedLLM and RouterBench, we use binary accuracy as the per-sample evaluation metric, meaning an LLM response is classified strictly as correct or incorrect. For MixInstruct, we employ exponentiated BARTScore [97] as the evaluation metric, following the approach in prior work [40, 39].

Table 6: Full breakdown of training and testing categories used in OOD experiments.

| Set | Categories |
|-----|-----------|
| Train (STEM) | asdiv, gsm8k, medmcqa, mathqa, piqa, logiqa, gpqa_main_cot_n_shot, gpqa_main_cot_zeroshot, gpqa_main_n_shot, gpqa_main_zeroshot, gpqa_diamond_cot_n_shot, gpqa_diamond_cot_zeroshot, gpqa_diamond_n_shot, gpqa_diamond_zeroshot, gpqa_extended_cot_n_shot, gpqa_extended_cot_zeroshot, gpqa_extended_n_shot, gpqa_extended_zeroshot, mmlu_college_medicine, mmlu_astronomy, mmlu_conceptual_physics, mmlu_college_computer_science, mmlu_college_biology, mmlu_electrical_engineering, mmlu_medical_genetics, mmlu_college_physics, mmlu_high_school_chemistry, mmlu_computer_security, mmlu_clinical_knowledge, mmlu_virology, mmlu_machine_learning, mmlu_college_mathematics, mmlu_elementary_mathematics, mmlu_professional_medicine, mmlu_college_chemistry, mmlu_high_school_biology, mmlu_anatomy, mmlu_high_school_statistics, mmlu_high_school_physics, mmlu_high_school_computer_science, mmlu_high_school_mathematics |
| Test (Non-STEM) | social_iqa, truthfulqa_mc1, mmlu_high_school_european_history, mmlu_us_foreign_policy, mmlu_high_school_microeconomics, mmlu_business_ethics, mmlu_public_relations, mmlu_jurisprudence, mmlu_nutrition, mmlu_high_school_world_history, mmlu_miscellaneous, mmlu_formal_logic, mmlu_management, mmlu_high_school_psychology, mmlu_high_school_government_and_politics, mmlu_high_school_geography, mmlu_world_religions, mmlu_international_law, mmlu_human_aging, mmlu_sociology, mmlu_professional_accounting, mmlu_prehistory, mmlu_logical_fallacies, mmlu_moral_disputes, mmlu_human_sexuality, mmlu_professional_psychology, mmlu_high_school_us_history, mmlu_high_school_macroeconomics, mmlu_abstract_algebra, mmlu_global_facts, mmlu_security_studies, mmlu_philosophy, mmlu_professional_law, mmlu_moral_scenarios, mmlu_marketing |

We evaluate each routing strategy using a deferral curve [40] which plots the average response quality against the total inference cost. Sweeping the routing penalty parameter $\lambda$ over the interval $\lambda \in [0, \lambda_{\max}]$ (refer to Equation (9)) traces the deferral curve. For the EmbedLLM and MixInstruct datasets, we define the cost of processing a prompt as the number of parameters in the LLM, serving as a proxy for computational resources and latency. In the case of RouterBench, we utilize the actual API call costs in USD, as provided in the dataset. Following [40] we employ evaluation metrics including Area Under the Deferral Curve (AUDC), Query-Normalized Cost (QNC), and peak accuracy. QNC is the minimum relative cost required to match the performance of the most accurate tested LLM.

**Training** We use a frozen `sentence-transformers/all-MiniLM-L6-v2`[70] model as the embedding backbone ($\Phi(x)$ in Section 3) across all experiments. Our trainable router component is a two-layer MLP, denoted as $g_\theta(.)$, which projects prompt embeddings into the expert descriptor space. We train our contrastive router on the training splits of each dataset, excluding the probe examples from RouterBench. Training is performed for 10 epochs using the AdamW optimizer with a batch size of 512 and a learning rate of $5 \times 10^{-4}$. For the cost spectrum loss (Equation (8)), we set the number of cost bands to $K = 5$ and the negative cost penalty to $\lambda = 0.1$. The hyperparameters for the linear schedule of band-specific temperatures (Equation (7)) are set as $\alpha = 0.25$ and $\tau_{\min} = 0.05$. All training and descriptor extraction are done on RTX6000Ada GPUs with ~48GB GPU memory.

## D.2 Results

### D.2.1 Out-of-Distribution Prompt Experiments

In the out-of-distribution (OOD) experiments, we divided the prompts in the EmbedLLM dataset into two challenging sets based on their categories: **STEM-related** (Science, Technology, Engineering, and Mathematics) and **Non-STEM-related** (covering Social sciences, Humanities, Arts, etc.). A detailed summary of the train and test categories for the OOD experiments is provided in Table 6. Our splitting yielded 18,193 out of 36,054 total training questions and 1,060 distinct test prompts. Training and testing is done on all available LLMs across both splits.

## D.3 Ablation of Cost Bands

We measured AUCD and peak accuracy with and without bands. Table 7 presents these results, showing having cost bands is indeed effective.

| Setting | AUCD | Peak Accuracy |
|---------|------|---------------|
| Without bands | 0.4574 | 0.482 |
| With bands | 0.4951 | 0.540 |

Table 7: Effect of banded cost temperatures on model performance

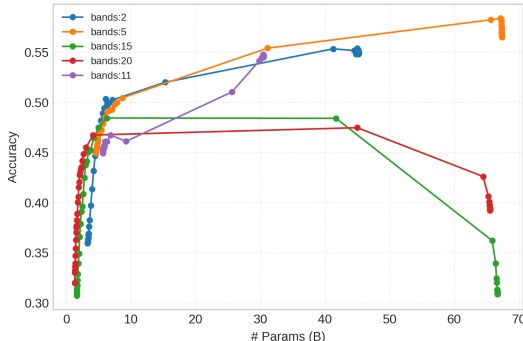

| Router (Bands) | AUDC ↑ | QNC ↓ | Peak ↑ |
|---|---|---|---|
| bands:2 | 0.5357 | 47.693 | 0.563 |
| bands:5 | 0.5480 | 66.057 | 0.590 |
| bands:11 | 0.5173 | 30.412 | 0.547 |
| bands:15 | 0.4460 | 13.474 | 0.496 |
| bands:20 | 0.4649 | 11.541 | 0.477 |

Table 8: Deferral-curve metrics across different numbers of cost bands.

Figure 7: Deferral-curve metrics across different numbers of cost bands.

| $K$ | AUDC | cost@max_acc |
|---|---|---|
| 1 | 0.5186 | 7.665 |
| 2 | 0.5294 | 7.759 |
| 4 | 0.5344 | 7.831 |
| 8 | 0.5378 | 7.940 |
| 16 | 0.5402 | 8.034 |

Table 9: Effect of Number of Nearest Neighbors ($K$) on AUDC and Average Cost

### D.3.1 Ablation of Number of Cost bands

Table 8 and Figure 7 show that performance varies with the number of cost bands. Using 5 bands yields the best results, with the highest AUDC (0.5480) and Peak accuracy (0.590), suggesting a good balance between flexibility and generalization. Fewer bands (e.g., 2) limit routing precision, while too many bands (e.g., 15 or 20) degrade performance, likely due to over-fragmentation and increased decision noise. This highlights the importance of tuning the number of bands to avoid both under- and overfitting.

### D.3.2 Ablation of the Number of Neighbors

Increasing $k$ (the number of ANN neighbors selected before cost-aware scoring) generally provides modest improvements before reaching a plateau. (The baseline UMR [40] also included an ablation on $k$ in a K-NN router.) Beyond a small $K$, the gains in accuracy or AUDC become minimal, while both latency and the likelihood of choosing unnecessarily expensive experts increase. We selected a default of $K = 4$ and it worked reasonably well so we did not do further hyperparameter tuning. We performed an ablation on a subset of `embedllm` prompts. The results are shown in Table 9 which are consistent with the trend observed in UMR.

### D.3.3 Ablation of Negative Cost Penalty

$\gamma$ is a soft deterrent against assigning probability mass to costly, wrong experts during training (it appears in the denominator of the band-softmax in the loss in Equation 8. We ablate this hyperparameter in Table 10. If too small, the router learns to over-consider expensive hub experts (cost creep). If too large, it over-penalizes cost and undertrains on valid high-cost positives, hurting hard prompts. We picked $0.2$ as it's a light regularizer: enough to push apart costly negatives first, but not so strong that it drowns the similarity signal for truly necessary expensive experts.

### D.3.4 Ablation of Band-Specific Temperature Slope

In the band-specific temperature schedule $\tau_k = \tau_{\min} + \alpha \bar{c}_k$ (Equation 7), the slope $\alpha$ sets how much flatter the softmax is in higher-cost bands. Increasing $\alpha$ raises $\tau_k$ for expensive bands, which flattens their per-band softmax over experts. This reduces gradient variance in those bands (where each query

| Router | AUDC | max_acc | cost@max_acc |
|---|---|---|---|
| $\gamma$=0 | 0.5518 | 0.5980 | 55.543 |
| $\gamma$=0.1 | 0.5566 | 0.5850 | 44.499 |
| $\gamma$=0.2 | 0.5526 | 0.5720 | 43.083 |
| $\gamma$=0.3 | 0.5272 | 0.5307 | 14.901 |
| $\gamma$=0.5 | 0.5129 | 0.5147 | 8.836 |

Table 10: Gamma ablation

| $\alpha$ | AUDC | max_acc | cost@max_acc |
|---|---|---|---|
| 0 | 0.5567 | 0.5930 | 64.292 |
| 0.1 | 0.5704 | 0.6033 | 55.680 |
| 0.25 | 0.5701 | 0.6003 | 48.360 |
| 0.4 | 0.5557 | 0.5657 | 32.854 |
| 0.5 | 0.5448 | 0.5517 | 29.175 |

Table 11: Ablation over the band slope $\alpha$ in $\tau_k = \tau_{\min} + \alpha\,\bar{c}_k$. Moderate $\alpha$ (0.1–0.25) maximizes AUDC and lowers the cost required to reach peak accuracy. Large $\alpha$ oversmooths high-cost bands and reduces peak accuracy.

has fewer suitable positive) mitigating collapse onto a single rare positive and preventing cost-creep during training.

See Table 11 for an ablation of this hyperparameter. As $\alpha$ increases from 0 to a moderate value (0.1–0.25), *AUDC* improves and the *cost@max_acc* drops, indicating we reach peak quality at lower cost, while *max_acc* remains comparable. For larger $\alpha$ ($\geq 0.4$), the high-cost bands become oversmoothed, weakening discrimination among expensive experts so *max_acc* and *AUDC* decline despite further cost reductions. This validates our choice of adopting a moderate setting (default $\alpha$=0.25, with 0.1 performing slightly better).

### D.3.5 Ablation of Cheapest Cost Band

$\tau_{\min}$ is the softmax temperature for the cheapest cost band in Equation 7. All other bands inherit $\tau_k = \tau_{\min} + \alpha\,\bar{c}_k$. A very small $\tau_{\min}$ makes the cheap-band softmax sharp (highly discriminative but prone to noisy, peaky gradients) while a larger $\tau_{\min}$ smooths the distribution, lowering variance but also blurring differences among cheap experts.

Table 12 ablates $\tau_{\min}$. Raising $\tau_{\min}$ from 0 to 0.02 increases AUDC and peak accuracy, showing that a touch of smoothing stabilizes learning without hurting discrimination. At $\tau_{\min} = 0.05$ we keep nearly the same AUDC while cutting the cost needed to achieve peak accuracy by $\approx 14\%$ (from 51.7 to 44.6). Pushing to $\tau_{\min} = 0.08$ oversmoothes the cheap band: accuracy at low cost rises slightly, but *max_acc* and AUDC both fall. Thus a moderate setting ($\tau_{\min} \approx 0.02$–0.05) offers the best efficiency–stability trade-off.

### D.4 Larger Encoder Size

We kept the router is deliberately small: a frozen sentence-transformer plus a 2-layer MLP. We performed an ablation where we increased the dimension of the middle layer in Table 13. As the router size increases, AUDC improves, but latency also increases.

| $\tau_{\min}$ | AUDC | max_acc | cost@max_acc |
|---|---|---|---|
| 0 | 0.5484 | 0.5877 | 51.232 |
| 0.02 | 0.5602 | 0.6037 | 51.704 |
| 0.05 | 0.5581 | 0.5883 | 44.633 |
| 0.08 | 0.5515 | 0.5707 | 38.395 |

Table 12: Ablation over the base temperature $\tau_{\min}$ (with $\alpha = 0.25$). Moderate values improve AUDC and lower the cost required for peak or near-peak accuracy. Very small or large values underperform.

| Router  | AUDC   |
|---------|--------|
| MLP-x1  | 0.5189 |
| MLP-x4  | 0.5384 |

Table 13: Effect of Router Size on AUDC

| Band Index | Count |
|------------|-------|
| 0          | 533   |
| 1          | 837   |
| 2          | 805   |
| 3          | 694   |
| 4          | 131   |

Table 14: Prompt Counts by Band Index

Broadly, future work can answer: How big must a model be to recognize a problem's difficulty even if it can't solve the problem itself? Future work can also study whether size and fine-tuning helps and whether RL on LLMs makes them able to purely as routers? Future work can also study different model sizes to pinpoint when difficulty awareness kicks in.

### D.5 Qualitative Insights and Interpretability of Routing

We performed an analysis on `embedllm`, which features a large pool of models with diverse costs. Note that these observed trends are specific to the dataset and may not generalize to other datasets or prompt pools with more challenging examples.

#### D.5.1 Selection Profiles

`Qwen/Qwen1.5-0.5B-Chat` (selected 91 times), `google/gemma-2b-it` (233), and `microsoft/phi-2` (292) are smaller experts that are selected frequently. Some expensive experts (e.g., `Qwen/Qwen-72B`, `ibivibiv/alpaca-dragon-72b-v1`) are rarely chosen, since a less costly correct expert typically exists in the dataset.

We also report per-expert selection rates by cost band in Table 14.

#### D.5.2 Routing Error Breakdown

We present a confusion-style breakdown of routing errors in Table 15

- Too cheap—the router selects a cheap but incorrect expert

- Too expensive—the router selects an unnecessarily costly expert, though correct

- Optimal—the router selects a correct and minimally costly expert

- No correct—no available expert produces a correct answer

| Outcome        | Count |
|----------------|-------|
| Too cheap      | 235   |
| Too expensive  | 586   |
| Optimal        | 773   |
| No correct     | 63    |

Table 15: Routing Outcome Breakdown

| Dataset | $\Delta_{\mathbf{AUDC}}$ | 95% CI | $p$ (bootstrap) | $c^\star$ | $n_{10}/n_{01}$ | $p$ (McNemar) |
|---|---|---|---|---|---|---|
| EMBEDLLM | +0.053 | [0.037, 0.069] | $2.0 \times 10^{-4}$ | 6.08 | 560/366 | $9.76 \times 10^{-11}$ |

Table 16: Paired significance versus the strongest baseline. $\Delta_{\text{AUDC}} = \text{AUDC}_{\text{CSCR}} - \text{AUDC}_{\text{UMR}}$ (area under the deferral curve. Higher is better). "95% CI" and the one-sided $p$ come from a paired bootstrap over prompts ($N = 3000$, $B = 5000$, $H_1 : \Delta > 0$). $c^\star$ is the matched budget used for McNemar. $n_{10}/n_{01}$ are discordant counts (CSCR correct / baseline correct), and "$p$ (McNemar)" is the one-sided exact binomial $p$ for CSCR > baseline at $c^\star$. CIs that exclude 0 and small $p$-values indicate a statistically significant improvement of CSCR.

## D.6 Statistical Significance

We perform full evaluation on Embedllm [105] using paired, prompt-level significance tests to concretely assess statistical significance. Specifically, we computed a paired bootstrap [22] (sampling 3,000 prompts with replacement, 5,000 times) to obtain a 95% confidence interval for $\Delta_{\text{AUDC}} = \text{AUDC}_{\text{CSCR}} - \text{AUDC}_{\text{UMR}}$ (UMR is the best baseline overall). We also report a one-sided p-value for the hypothesis $\Delta > 0$. Additionally, we ran McNemar's [57] test at a matched budget (using the median of the combined cost grids) to compare per-prompt wins and losses at equivalent operating cost. These tests quantify uncertainty over the test prompts.

$\Delta$AUDC is positive with CIs that exclude zero, and McNemar shows win rates above 0.5 with very small p-values at the matched budget. In other words: CSCR's deferral curve encloses more area (higher accuracy at the same or lower cost on average), and at a fixed budget it wins on more prompts than it loses. This complements the Pareto-frontier plots: the gains are not an artifact of a single operating point or random variation, but hold paired, prompt by prompt. The paired test establishes statistical significance.

