# OpenReview forum: "Cost-Aware Contrastive Routing for LLMs"
_NeurIPS.cc/2025/Conference — NeurIPS 2025 spotlight_

### Official Review · Reviewer_9rPU · 2025-06-06

**Clarity:** 2
**Significance:** 3
**Originality:** 3
**Rating:** 5
**Confidence:** 3

**Summary:**

This paper introduces Cost-Spectrum Contrastive Routing (CSCR), a lightweight framework for dynamically selecting the most cost-effective LLM from a diverse model pool to handle a given user prompt. It works by embedding both prompts and LLM experts into a shared embedding space, and selecting the most suitable model for a given user prompt based on the embedding similarity. The authors give two methods to obtain model embeddings (Fingerprints): Logit–Footprint Descriptors for Transparent LLMs, and Perplexity Fingerprints for Black-Box LLMs. A query encoder is trained to obtain the prompt embedding via a contrastive training objective. At inference time, routing is reduced to a fast k-NN lookup in this learned embedding space, and the final expert is selected based on both the embedding similarity and cost. Experiments show that CSCR consistently outperforms other routers, while also demonstrating strong generalization to unseen LLMs and out-of-distribution prompts.

**Questions:**

None

**Ethical Concerns:**

["NO or VERY MINOR ethics concerns only"]

**Final Justification:**

This response addresses my concern, so I'll keep my positive rating.

**Limitations:**

yes

**Quality:**

3

**Strengths And Weaknesses:**

Strengths
1. The proposed method is novel and intuitive.
2. CSCR outperforms baselines mehods, while being lightweight, cost-aware, and generalizable.
3. The evaluation is comprehensive.

Weaknesses
1. ​​The presentation could be improved. For instance, in Section 3.1.2, the transition from Equation 3 to Equation 5 isn't sufficiently clear. The rationale behind this approximation needs better justification or explanation.

---

> ### Author Rebuttal · Authors · 2025-07-31
>
> We sincerely thank the reviewer for the positive assessment and for pointing out the presentation gap between Eq. 3 and Eq 5.
> Below we state the missing bridge explicitly and justify the approximation.
> ***
> # The step from Eq. (3) to Eq. (5).
> Eq. (3) defines a per-prompt token NLL that requires access to an expert’s next-token distribution $p_h(\cdot \mid \cdot)$.
> For *API-only (black-box) experts*, logits/probabilities are not exposed, so Eq. (3) is not computable.
> Our remedy is to
> 1. let the API expert $h$ produce a deterministic continuation $\hat{y}_h(x)$ for prompt $x$ (greedy decoding), and
> 2. evaluate that sequence under a single shared, public scorer $p_S$.
>
> This yields Eq. (5), a *pseudo-perplexity*:
>
> $$
> \tilde{\ell}_h(x) = -\frac{1}{|\hat{y}_h(x)|} \sum _{t = 1} ^{|\hat{y}_h(x)|}  \log p_S ( \hat{y} _{h,t} | \hat{y} _{h,<t} )
> $$
>
> which we then normalize (similar to Eq. 4) and use as the fingerprint coordinate(s) for black-box experts. If $\hat y_h$ is a typical (high-probability) output of $h$ (i.e., $\hat y_h \sim p_h$) then averaging the pseudo-perplexity $\tilde{\ell} _h(x)  = -\tfrac{1}{|\hat{y}_h(x)|} \sum _{t} \log p_S ( \hat{y} _{h,t} | \hat{y} _{h,<t})$
> over many prompts/tokens is equivalent to taking an expectation over $y \sim p_h$:
>
> $$
> \mathbb{E}_{y\sim p_h}\big[-\log p_S(y)\big]
> = H(p_h,p_S)
> = H(p_h) + \mathrm{KL} \big(p_h || p_S \big).
> $$
>
> Here $H(p_h,p_S)$ is the *cross-entropy* of $p_h$ with respect to $p_S$, which decomposes into
> the entropy of $h$’s own distribution $H(p_h)$ and its divergence from the scorer $\mathrm{KL}(p_h\|p_S)$.
>
>
> Because $p_S$ is fixed for all experts, $H(p_h,p_S)$ is a stable, model-specific quantity that makes descriptors
> comparable across experts (same yardstick). It is not the true NLL under $p_h$, but it preserves differences between experts
> via $H(p_h)$ and their mismatch to $p_S$ via $\mathrm{KL}(p_h\|p_S)$. In practice we use deterministic (greedy) decoding to
> reduce variance; averaging over many prompts/tokens makes the empirical $\tilde{\ell}_h(x)$ closely track the expectation above.
> Figure 2 in the paper empirically validates this argument.
>
> We will make sure we add an explanation so the transition from Eq. 3 to Eq. 5 is smooth.
>
> ***
> We thank you for your feedback again. This improves the clarity of our work. We have also conducted additional ablation experiments which we invite you to read in our response to reviewer PkxW.

---

> > ### Comment · Reviewer_9rPU · 2025-08-01
> >
> > Thank you for your clarifications and explanations!

---

> > > ### Author Response · Authors · 2025-08-02
> > >
> > > Thank you again for your thoughtful review and for engaging with our responses. Your feedback was genuinely helpful in strengthening our work.

---

### Official Review · Reviewer_PkxW · 2025-06-08

**Clarity:** 2
**Significance:** 2
**Originality:** 3
**Rating:** 4
**Confidence:** 2

**Summary:**

This paper proposes cost-spectrum contrastive routing (CSCR), which improves inference efficiency by routing simpler queries to smaller models, while maintaining comparable accuracy. CSCR first involves mapping every LLM to a model fingerprint vector (either logit-based for open LMs, or perplexity-based for API-only LLMs). Second, a contrastive router is trained to learn a shared embedding space where query vectors are pushed closer to right-sized experts, using a modification of the InfoNCE loss. At inference time, a given prompt is routed to the most cost-effective expert. CSCR is a lightweight, 2-layer MLP and is tested on the baselines EmbedLLM, MixInstruct, and RouterBench, achieving up to 25% higher accuracy-cost efficiency.

**Questions:**

Questions:
- I am not sure if I’ve missed this anywhere, but is there any ablation of the k used in Equation 9? In general, I'm curious about the selection of hyper-parameters and further ablation studies would be helpful.
- What happens if you ablate the encoder used to learn the shared embedding space? Would a larger encoder improve performance?

Typos:
- L34: Routing → routing
- L148: classical -> classic

**Ethical Concerns:**

["NO or VERY MINOR ethics concerns only"]

**Final Justification:**

As stated below, I've elected to keep my score. The authors have responded to all my points sufficiently and I think this paper would be a good addition to the venue.

**Limitations:**

Yes

**Quality:**

2

**Strengths And Weaknesses:**

Strengths:
- This is a very relevant and timely problem; as LM inference with many different options and pricing schemas is very common these days, so figuring out the optimal tradeoff between cost and accuracy is important.
- Method relies on a small, efficient encoder and seems efficient at run time.
- The authors introduce a new InfoNCE loss and ablate it in 4.3.2, finding it to be more cost-effective.

Weaknesses:
- Are the main results (Tables 1, 2, 3) statistically significant? I know that the authors cited computational cost as why they do not include statistical significance (which is not a good reason IMO given that even statistical estimation is typically not prohibitive), but some reported metrics (e.g., AUDC) look very close together, which makes me hesitant to draw any definitive conclusions. Further, this is a little confusing to me since the low cost and efficiency seems to be the main advantage of the method.
- I wonder if there are any qualitative insights to be made about the router decisions (i.e., if interpretability could be a potential advantage). For instance, are there any LMs that punch above their weight class and tend to be chosen more often despite being cheaper to serve? Similarly, are there any more expensive LMs that are almost never selected?
- Writing is unclear and confusing in many places, and the manuscript may benefit from a more thorough proofreading. In particular, the introduction has lots of gratuitous bolding and italics, which may be distracting to the reader.

---

> ### Author Rebuttal · Authors · 2025-07-31
>
> We sincerely thank the reviewer for the positive assessment, the thoughtful feedback and actionable suggestions.
> ***
> # 1. On statistical significance
> Our router consistently outperforms all baselines: AUDC is higher on every dataset, and the deferral curves align with or extend the empirical accuracy–cost Pareto frontier, rather than simply trading wins and losses across different regimes. Furthermore, AUDC is an average metric across the full deferral curve, not just a single accuracy point on a split dataset. If the reviewer is referencing a specific result, could they please clarify which values?
>
> We had considered running robustness experiments with different probe sets to report mean and standard deviation, but this would require repeating each experiment at least three times, which is computationally prohibitive. That is what we referred to in the checklist.
>
> As requested by the reviewer, we re-ran the full evaluation on EmbedLLM using paired, prompt-level significance tests to concretely assess statistical significance. Specifically, we computed a paired bootstrap [1] to obtain a $95\\%$ confidence interval for
> $\Delta_{AUDC}$ (Difference between AUDC of CSCR and AUDC of UMR as the best baseline overall).
> We also report a one-sided p-value for the hypothesis $\Delta > 0$. Additionally, we ran McNemar’s test [2] at a matched budget (using the median of the combined cost grids) to compare per-prompt wins and losses at equivalent operating cost. These tests quantify uncertainty over the test prompts, which we believe matches the reviewer’s intended notion of significance.
>
> |Dataset|$\Delta_{AUDC}$|$95\\%$ CI|$p (\text{bootstrap})$|$c^\star$ | $n_{10} \/ n_{01}$|$p (\text{McNemar})$|
> |-|-|-|-|-|-|-|
> | EmbedLLM|+0.053| [0.037,0.069] |2.0×10⁻⁴|6.08|560/366|9.76×10⁻¹¹|
>
>  $95\\%$ Confidence Interval and the one-sided $p$ come from a paired bootstrap over prompts ($N=3000$, $B=5000$; $H_1 : \Delta > 0$).  $c^\star$ is the matched budget used for McNemar.
> $n_{10}/n_{01}$ are discordant counts (CSCR correct / UMR correct), and “$p$ (McNemar)” is the one-sided exact binomial $p$ for CSCR $>$ baseline at $c^\star$.
>
> $\Delta_{AUDC}$ is positive with CIs that exclude zero, and McNemar shows win rates above 0.5 with very small p-values at the matched budget. In other words: CSCR’s deferral curve encloses more area (higher accuracy at the same or lower cost on average), and at a fixed budget it wins on more prompts than it loses. This complements the Pareto-frontier plots: the gains are not an artifact of a single operating point or random variation, but hold paired, prompt by prompt. The establishes statistical significance.
>
> # 2. Writing clarity
> We will remove the bold/italics in the introduction and other sections, clarify writing, and do another full proofread. We appreciate the pointers and have fixed the two typos the reviewer noted.
>
> # 3. More Ablations
> Most of our choices for the contrastive loss were based on our past experience training contrastive models. We conducted additional ablation studies during the rebuttal period as per the reviewer's request.
>
> ## 3.1 $K$ in Eq. (9)
> We selected a default of $K=4$  and it worked reasonably well so we did not do further hyperparameter tuning. However, we agree with the reviewer that an ablation would be useful, so we performed one on a subset of EmbedLLM prompts.
>
> |K|AUDC|cost@max_acc|
> |-|-|-|
> |1|0.5186|7.665|
> |2|0.5294|7.759|
> |4|0.5344|7.831|
> |8|0.5378|7.940|
> |16|0.5402|8.034|
>
> Increasing $K$ generally provides modest improvements before reaching a plateau. Beyond a small $K$, the gains in accuracy or AUDC become minimal, while both latency and the likelihood of choosing unnecessarily expensive experts increase. The baseline UMR also included an ablation on $K$ in a K-NN router. The trends are similar.
>
> ## 3.2 Banded Costs
> We measured AUCD and peak accuracy with and without bands to validate having banded cost.
>
> |Setting|AUCD|Peak Accuracy|
> |-|-|-|
> |W/ bands|0.4574|0.482|
> |W/O bands|0.4951|0.540|
>
> ## 3.3  $\gamma$ in Eq. 8
> $\gamma$ is a soft deterrent against assigning probability mass to costly, wrong experts during training (the denominator of the band-softmax in Eq. 8). If too small, the router learns to over-consider expensive hub experts. If too large, it over-penalizes cost and undertrains on valid high-cost positives, hurting hard prompts. We picked 0.2 as it’s a light regularizer: enough to push apart costly negatives first, but not so strong that it drowns the similarity signal for truly necessary expensive experts.
>
> | $\gamma$| AUDC   | max_acc | cost@max_acc |
> |--|-|-|-|
> |0| 0.5518 | 0.5980  | 55.543  |
> | 0.1| 0.5566 | 0.5850  | 44.499 |
> | 0.2 | 0.5526 | 0.5720  | 43.083 |
> | 0.3 | 0.5272 | 0.5307  | 14.901 |
> |0.5| 0.5129 | 0.5147  | 8.836  |
>
> ## 3.4 $\alpha$
>
> In the band-specific temperature schedule $\tau_k=\tau_{\min}+\alpha\,\bar c_k$, the slope $\alpha$ sets how much flatter the softmax is in higher-cost bands. Increasing $\alpha$ raises $\tau_k$ for expensive bands, which flattens their per-band softmax over experts. This reduces gradient variance in those bands (where each query has fewer suitable positive) mitigating collapse onto a single rare positive and preventing cost-creep during training.
>
> | $\alpha$ | AUDC   | max_acc | cost@max_acc |
> |--|-|--|--|
> | 0     | 0.5567 | 0.5930  | 64.292       |
> | 0.1   | 0.5704 | 0.6033  | 55.680       |
> | 0.25  | 0.5701 | 0.6003  | 48.360       |
> | 0.4   | 0.5557 | 0.5657  | 32.854       |
> | 0.5   | 0.5448 | 0.5517  | 29.175       |
>
> As $\alpha$ increases from $0$ to a moderate value ($0.1$–$0.25$), *AUDC* improves and the *cost@max\_acc* drops, indicating we reach peak quality at lower cost, while *max\_acc* remains comparable. For larger $(\alpha \ge 0.4$), the high-cost bands become oversmoothed, weakening discrimination among expensive experts; *max\_acc* and *AUDC* decline despite further cost reductions. This validates our choice of adopting a moderate setting (default $\alpha=0.25$, with $0.1$ performing slightly better).
>
> ## 3.5 $\tau_{\text{min}}$
> $\tau_{\min}$ is the softmax temperature for the cheapest cost band; all other bands inherit $\tau_k=\tau_{\min}+\alpha\,\bar c_k$.  A very small $\tau_{\min}$ makes the cheap-band softmax sharp (highly discriminative but prone to noisy, peaky gradients) while a larger $\tau_{\min}$ smooths the distribution, lowering variance but also blurring differences among cheap experts.
>
> | $\tau_{\text{min}}$ | AUDC   | max_acc | cost@max_acc |
> |-|-|-|-|
> | 0 | 0.5484 | 0.5877 | 51.232  |
> | 0.02  | 0.5602 | 0.6037 | 51.704 |
> | 0.05  | 0.5581 | 0.5883 | 44.633 |
> | 0.08  | 0.5515 | 0.5707 | 38.395 |
>
> Raising $\tau_{\min}$ from $0$ to $0.02$ increases AUDC and peak accuracy, showing that a touch of smoothing stabilizes learning without hurting discrimination. At $\tau_{\min}=0.05$ we keep nearly the same AUDC while cutting the cost needed to achieve peak accuracy by $\approx14\\%$ (from 51.7 to 44.6).  Pushing to $\tau_{\min}=0.08$ oversmoothes the cheap band: accuracy at low cost rises slightly, but max\_ac and AUDC both fall.  Thus a moderate setting ($\tau_{\min} \approx 0.02$–$0.05$) offers the best efficiency–stability trade-off.
>
> # 4. Q2: What if we ablate or enlarge the encoder?
> We’re glad you asked as this is exactly the direction we’re exploring next.
> In this work we deliberately kept the router small: a frozen sentence-transformer plus a 2-layer MLP. We have added the following ablation where we increased the dimension of the middle layer of the MLP. As the router size increases, AUDC improves, but latency also increases.
>
> | Router | AUDC |
> |-|-|
> | MLP-x1 | 0.5189 |
> | MLP-x4 | 0.5384 |
>
> We’re now testing larger routers  by replacing the frozen query encoder with a larger LLM to see whether richer semantics improve difficulty judgment. More broadly, we’re excited to answer: How big must a model be to recognize a problem’s difficulty even if it can’t solve the problem itself? We want to study whether size and fine-tuning helps; we’re exploring whether RL on LLMs makes them able to purely as routers and sweep model size to pinpoint when difficulty awareness kicks in.
>
> # 5. Qualitative Insights and Interpretability
> We agree that interpretability is extremely valuable. We performed an analysis on embedllm, which features a large pool of models with diverse costs. Note that these observed trends are specific to the dataset and may not generalize to other datasets or prompt pools with more challenging examples.
>
> ## 5.1 Selection Profiles
> $\texttt{Qwen/Qwen1.5-0.5B-Chat}$ (selected 91 times), $\texttt{google/gemma-2b-it}$ (233), and $\texttt{microsoft/phi-2}$ (292)—are smaller experts that are selected frequently. Some expensive experts (e.g., $\texttt{Qwen/Qwen-72B}$, $\texttt{ibivibiv/alpaca-dragon-72b-v1}$) are rarely chosen, since a less costly correct expert typically exists in the dataset.
>
> We also report per-expert selection rates by cost band.
>
> | Band Index | Count |
> |-|-|
> | 0 | 533 |
> | 1 | 837 |
> | 2 | 805 |
> | 3 | 694 |
> | 4 | 131 |
>
> ## 5.2 Taxonomy of Errors
> We present a confusion-style breakdown of routing errors:
> 1. Too cheap—the router selects a cheap but incorrect expert
> 2. Too expensive—the router selects an unnecessarily costly expert, though correct
> 3. Optimal—the router selects a correct and minimally costly expert
> 4. No correct—no available expert produces a correct answer
>
> | Outcome | Count |
> |-|-|
> | Too cheap | 235 |
> | Too expensive | 586   |
> | Optimal | 773 |
> | No correct | 63|
>
> We will add more comprehensive analysis and charts to the Appendix.
>
> # 6. References
> [1] An Introduction to the Bootstrap, Efron and Tibshirani, 1994
> [2] Note on the sampling error of the difference between correlated proportions or percentages, McNemar, 1947
>
> ***
> We truly appreciate the reviewer’s suggestions; the above additions directly address significance, interpretability, and clarity or our work.

---

> > ### Comment · Reviewer_PkxW · 2025-08-02
> >
> > Thank you for the explanations and the additional experiments! I will retain my positive score of 4. Best of luck!

---

> > > ### Author Response · Authors · 2025-08-02
> > >
> > > We sincerely thank you for carefully considering our rebuttal. The insightful comments and engagement improved our paper.

---

### Official Review · Reviewer_7VYj · 2025-07-04

**Clarity:** 4
**Significance:** 4
**Originality:** 4
**Rating:** 5
**Confidence:** 4

**Summary:**

This paper introduces Cost-Spectrum Contrastive Routing (CSCR), a framework for cost-aware routing across pools of large language models (LLMs). ​ CSCR addresses limitations in existing routing methods, such as reliance on expensive profiling, inefficiency, and lack of adaptability to dynamic model pools.

**Questions:**

The paper is quite strong and very well-written. The related work section is extensive, crisp and very detailed. I have a few questions that I have provided below:

1. In equation 2, what is the logic of taking the K most frequent tokens? If K is not too large, then they will most likely correspond to trivial tokens such as a, an, the etc - isn't that correct? Should we use something more like tf-idf to select the tokens? Or do you think that not having a common set of tokens across the models will lead to an embedding that will not work. Can you please clear up this confusion?

2. How can dense human annotations further help the algorithm? In some cases, for a small number of datapoints, detailed human feedback might be available - how can it be further helpful?

3. Please expand on L167 further - why do smoother gradients help and how do they results from assigning band-specific temperatures? It is not clear to me why gradients are smoother because of the clustering. What happens if the gradients are not smooth for the hard examples?

**Ethical Concerns:**

["NO or VERY MINOR ethics concerns only"]

**Limitations:**

No limitations to the best of my understanding

**Quality:**

4

**Strengths And Weaknesses:**

Strengths: The paper is exceptionally well-written with abundant explanations and intuitions guiding the reader. The problem is clearly stated and intuitions behind the solutions and its advantages are clearly highlighted

No such weaknesses

---

> ### Author Rebuttal · Authors · 2025-07-31
>
> We sincerely thank the reviewer for the careful read, the positive assessment, and the great questions. We address the questions below and will incorporate the suggested clarifications into the paper.
> ***
> # 1 why take the $K$ most frequent tokens? Will they be trivial? Should we use TF-IDF instead?
>
> In short, we use the most frequent tokens so the basis is shared and stable: they appear in all models, give low-noise estimates with few probes, and make calibration comparable across experts.  They give less noisy estimates because they get non-negligible probability across many contexts, so their averaged log-probs vary less than rare or OOV tokens. In all experiments, we set $K=256$ and $T=10$ (Appendix C1), which is large enough that the basis isn’t dominated by a few function words. We will add a short paragraph to the paper stating this rationale explicitly.
>
> ## 1.1 Are frequent tokens "trivial"?
> These tokens aren’t used for their meaning. They’re probes of each model’s output behavior. Even common words get scored differently across models (temperature, punctuation/number handling, style). By averaging over many prompts and steps, the descriptor captures overall model behavior, not any single word’s semantics. Also, a recent work shows that LLMs exhibit stable, word-level *idiosyncrasies* (as the authors call them) that enable near-perfect model attribution using only the first few generated tokens (even after paraphrasing or translation), implying that common tokens still provide discriminative signals about a model’s predictive calibration [1].
>
> ## 1.2 Shared token set vs. per-model selection.
> A shared basis gives all descriptors a common coordinate system. If each model used a different token set, cosine distances would mix basis changes with true behavior, hurting comparability. It would also require computing many more probes to align per-model bases that are different across models.
>
> ## 1.3. Beyond raw frequency.
> We deliberately kept the descriptors simple to isolate and quantify the contrastive router’s contribution. We agree that frequency is a pragmatic, not necessarily optimal, choice. Two variants that we considered and could be explored are: (i) TF–IDF-weighted selection over the probe corpus, just like the reviewer suggested. (ii) picking tokens with the largest across-model log-prob variance. These can be dropped into Eq. (2) without changing downstream training or inference.
>
> If the reviewer believes that ablating this choice would strengthen the paper, we are happy to include it.
>
> # 2. How could dense human annotations help?
> Great question. We considered using human annotations but intentionally avoided them: model pools change quickly, so adding/replacing experts would require fresh labels that are costly and often unavailable. Instead, we train with sparse correctness and cost signals, which remain portable across experts. If dense feedback is available, it could help in several ways:
> * **Positive sets $P(i)$ with preference structure.** Replace binary “correct expert” labels with pairwise preferences (cheap‐and‐good $\\succ$ expensive‐and‐similar $\\succ$ clearly wrong), yielding band-aware positives and margin constraints. This can be implemented by expanding $P(i)$ and adding a lightweight pairwise ranking (DPO-style) regularizer within each cost band.
> * **Difficulty-aware reweighting.** Use human “difficulty” scores to upweight rare/hard prompts when computing the contrastive loss, especially in higher cost bands. this could balance the effective sample sizes across easy vs. hard (and cheap vs. expensive band) cases so the gradient isn’t dominated by the abundant, easy examples.
> * **Band calibration.** We can ask users how much quality they’re willing to trade for a lower cost, then use that to set the cost bands and the penalty for picking more expensive models, so the router’s choices match what users actually prefer.
>
> We will add a brief paragraph in Appendix describing these straightforward extensions.
>
>
> # 3 Expanding L167: why do band-specific temperatures yield smoother gradients? What if gradients are not smooth on hard examples?
> Thank you for catching this. We originally included a full explanation in the paper (lines 183-186), but it was accidentally removed during our edits for space and wasn’t moved to the appendix.
>
> Most prompts in everyday interactions (and in our datasets) can be handled by cheaper models; plus there are usually fewer very expensive experts overall. These expensive experts are only needed for a small fraction of hard prompts, so within those high-cost bands there are fewer suitable positives per query.  With few positives, the similarity distribution becomes very peaky.
>
> *Effect on gradients:* Increasing $\tau_k$ makes the softmax less concentrated, so (i) gradient magnitudes shrink roughly like  $1/\tau_k$ and (ii) directions average over more positives, reducing variance and preventing updates from collapsing onto a single rare positive, especially in the high-cost bands.
>
> *If gradients are not smooth for hard examples.* Without band-specific temperatures, we observed the router can have unstable training (possibly due to oscillatory updates on hard prompts) and slower convergence. It also performs worse (please see the additional ablations in our response to Reviewer PkxW).
>
> # 4. References
> [1] Idiosyncrasies in Large Language Models, Sun et al., ICML 2025
>
> ***
> This concludes our response to the reviewer’s question, but we invite them to read section 3 of our response to Reviewer  PkxW for additional ablation experiments including ablations on cost spectrum hyperparameters.  We hope these clarifications address the reviewer’s questions. We truly appreciate the insightful questions as they help increase the clarity of our work.

---

### Decision · Program_Chairs · 2025-09-17

**Decision:**

Accept (spotlight)

**Comment:**

This paper introduces Cost-Spectrum Contrastive Routing (CSCR), a framework for cost-aware routing across LLM pools that uses compact model descriptors and a contrastive objective incorporating inference cost. The approach is technically sound, lightweight, and achieves consistent improvements in accuracy–cost efficiency across multiple benchmarks, with demonstrated robustness to unseen models and out-of-distribution prompts. Reviewers noted strengths in novelty, clarity, and empirical validation, while raising concerns about statistical significance, interpretability, and presentation. The authors addressed these issues through additional experiments, analyses, and revisions during the rebuttal, which resolved the main concerns. Overall, the contribution is solid, timely, and practically relevant, and the AC recommends spotlight acceptance.